# Electric Fields Regulate In Vitro Surface Phosphatidylserine Exposure of Cancer Cells via a Calcium-Dependent Pathway

**DOI:** 10.3390/biomedicines11020466

**Published:** 2023-02-06

**Authors:** Ahmet Kaynak, Kombo F. N’Guessan, Priyankaben H. Patel, Jing-Huei Lee, Andrei B. Kogan, Daria A. Narmoneva, Xiaoyang Qi

**Affiliations:** 1Department of Biomedical Engineering, University of Cincinnati, Cincinnati, OH 45221, USA; 2Division of Hematology and Oncology, Department of Internal Medicine, University of Cincinnati College of Medicine, Cincinnati, OH 45267, USA; 3Department of Pathology and Laboratory Medicine, University of Cincinnati College of Medicine, Cincinnati, OH 45267, USA; 4Department of Biomedical Sciences, University of Cincinnati, Cincinnati, OH 45221, USA; 5Department of Physics, University of Cincinnati, Cincinnati, OH 45221, USA

**Keywords:** electric field, cancer biomarker, surface phosphatidylserine exposure, calcium modulation, p38 MAPK-actin pathway, enhanced cancer therapy

## Abstract

Cancer is the second leading cause of death worldwide after heart disease. The current treatment options to fight cancer are limited, and there is a critical need for better treatment strategies. During the last several decades, several electric field (EF)-based approaches for anti-cancer therapies have been introduced, such as electroporation and tumor-treating fields; still, they are far from optimal due to their invasive nature, limited efficacy and significant side effects. In this study, we developed a non-contact EF stimulation system to investigate the in vitro effects of a novel EF modality on cancer biomarkers in normal (human astrocytes, human pancreatic ductal epithelial -HDPE-cells) and cancer cell lines (glioblastoma U87-GBM, human pancreatic cancer cfPac-1, and MiaPaCa-2). Our results demonstrate that this EF modality can successfully modulate an important cancer cell biomarker-cell surface phosphatidylserine (PS). Our results further suggest that moderate, but not low, amplitude EF induces p38 mitogen-activated protein kinase (MAPK), actin polymerization, and cell cycle arrest in cancer cell lines. Based on our results, we propose a mechanism for EF-mediated PS exposure in cancer cells, where the magnitude of induced EF on the cell surface can differentially regulate intracellular calcium (Ca^2+^) levels, thereby modulating surface PS exposure.

## 1. Introduction

Following a seminal study by J. Teiessie et al., which reported the transient formation of membrane pores and phospholipid exposure in cells after exposing cancer cells to pulse-field electroporation [1], the application of electric field (EF) for cancer treatment has been the focus of significant research efforts. Reversible pore formation, also known as reversible electroporation, involves the breakdown of the bilayer structure of the cell membrane. Reversible electroporation has been used in preclinical and clinical studies with a variety of aims, including the induction of cytoskeleton instability [2], delivering anti-cancer agents or molecules into cancer cells [3,4], and the transfection of cells [5]. Ca^2+^-mediated electroporation is a new anti-cancer therapy approach currently being investigated in a phase II clinical trial [6]. It aims to facilitate intra-tumoral delivery of Ca^2+^ by reversible electroporation [7]. The amplification of Ca^2+^ intake with electroporation induces the severe depletion of ATP within the cells, leading to cancer cell death [8]. In addition to electroporation, the noninvasive application of alternating EF with intermediate amplitude is emerging as an anti-cancer treatment used in the clinic [9].

Tumor treating field (TTFields) used in combination with temozolomide was approved by the FDA in 2011 for the treatment of newly diagnosed and recurrent glioblastoma (GBM) [10]. There are currently several ongoing clinical trials investigating the application of TTFields in combination with chemo/radiation or immunotherapy for the treatment of a variety of cancers, including brain metastasis, lung small cell carcinoma [11], pancreatic cancer [12], mesothelioma [13], and ovarian cancer [14].

Although a series of breakthroughs on the effects of EFs on tumor cells have been made, clinically available modalities are still limited to electroporation and TTField for cancer treatment [15,16]. The discovery of novel EF modalities for the treatment of cancers would be beneficial.

Targeting cancer biomarkers and tumor-associated antigens have been a promising approach for cancer treatment and imaging over the years. These biomarkers include glycans, lipids, and cancer surface antigens. Our growing understanding of the basic science of cancer and technological improvements have led to the discovery of novel biomarkers that can be used to target cancer cells [17]. Despite the mounting interest in biomarker-driven cancer therapy, their role in EF treatment modalities remains largely unknown. Phosphatidylserine (PS) is a negatively charged phospholipid, mainly expressed in the inner leaflet of the cell membrane of healthy cells. However, in distressed cells and cancer cells, it is mostly expressed on the outer cell membrane [18]. PS transfer between the inner and outer leaflets of the cell membrane is mediated by flippase enzymes, which are inhibited by intracellular Ca^2+^ [18]. PS plays a critical role by serving as a marker for the clearance of apoptotic cells by macrophages [19]. Our group and others have shown that PS targeting is a promising approach for treating several cancer types, including pancreatic cancer, lung cancer, glioblastoma [20,21,22], and other maladies, such as Gaucher and Pompe diseases [23,24].

PS cell surface exposure is heterogeneous within a given population of cancer cells [18]. Our group has shown that the cancer cells with higher PS expression on the cell surface are more sensitive to PS-targeting treatment such as saposin C-coupled dioleoylphosphatidylserine (SapC-DOPS) nanovesicles [25,26]. Conversely, it has been shown that cancer cells with lower cell surface PS expression are more sensitive to chemotherapy and radiation [20,27]. In particular, cancer cells with high or low surface PS exposure can be selectively killed by these treatments. These results suggest that altering PS surface exposure of the cancer cells might sensitize them to SapC-DOPS nanovesicles or chemo/radiation therapy. Our group has also shown that the anti-cancer drug gemcitabine and radiation preferentially target cells in the G1 phase of the cell cycle, when cells express lower levels of surface PS, as compared to cells in the G2/M phase of the cell cycle [20,27]. In contrast, SapC-DOPS preferentially targets cells in the G2/M phase cells, which have high surface PS [20].

In this study, we demonstrate that external direct current EF generated using a parallel plate capacitor device can be used to differentially modulate surface PS expression on cancer cells depending on the EF amplitude. Our mechanistic studies show that EF-mediated regulation of surface PS expression of cancer cells occurs via changes in intracellular Ca^2+^ and actin polymerization in a time-dependent manner. We further demonstrate that EF regulates intracellular Ca^2+^ in cancer cells. Moreover, moderate amplitude leads to the activation of p38 mitogen-activated protein kinase (MAPK) through the phosphorylation of p38 MAPK and causes cell cycle arrest in the G2/M phase in cancer cells. On the other hand, low amplitude inhibits p38 MAPK activation. Overall, this work suggests a novel direction for the controlled manipulation of PS and targeted cancer therapies.

## 2. Materials and Methods

### 2.1. Cell Culture

The particular cell lines were selected for the experiments according to their PS exposure and voltage-gated calcium channel expression profile. It has been shown that voltage-gated channels are upregulated in both GBM and PDAC [28,29,30]. Therefore, we used U87-GBM and cfPac-1 cancer cells to detect the effects of EF on voltage-gated calcium channel activity and regulation of downstream pathways. Also, U87-GBM and cfPac-1 cell lines have a higher PS exposure level as compared to other cancer cells, which may result in a stronger (better detectable) response to low and moderate EF in terms of PS exposure change, i.e., if the PS exposure of cancer cells is low, EF may not decrease PS exposure significantly because it is already low. Glioblastoma cell line (U87-GBM) and human pancreatic cancer cell lines (cfPac-1 cells and MiaPaCa-2) were grown in DMEM (Corning, Manassas, VA, USA) supplemented with 10% FBS (Gibco, Grand Island, NY, USA), and 1% penicillin/streptomycin (Corning, Manassas, VA, USA). The MiaPaCa-2 cell line is an established human pancreatic ductal adenocarcinoma cell line [31] and is commonly used in stress-related research [32,33]. Human astrocyte cells were grown in an astrocyte medium (ScienCell, Carlsbad, CA, USA) supplemented with the provided growth factors, FBS, and antibiotics (Gibco, Grand Island, NY, USA. Human pancreatic ductal epithelial (HPDE) cells were grown in a keratinocyte medium (Gibco, Grand Island, NY, USA) with the provided growth factor supplements, FBS, and antibiotics. All cells were cultured in a 5% CO_2_ incubator at 37 °C. All cells were routinely confirmed to be negative for mycoplasma contamination using MP Biomedicals™ Myco-Sniff-Rapid™ Mycoplasma Luciferase Detection Kit (MP Biomedicals, Santa Ana, CA, USA).

### 2.2. In Vitro Electric Field Exposure Set-Up

A custom setup was built to allow cells exposure to EF using capacitive coupling (Figure 1A). This method used a parallel plate capacitor (two plates 135 mm × 128 mm and a distance of 26 mm between the plates) connected to a voltage source (Pasco, model SF-9586, Roseville, CA, USA) and assembled in the cell culture incubator. The cells were seeded in a petri dish filled with cell culture media, and the petri dish was placed between the capacitor plates so that cells did not have any contact with the electrodes. For these experiments, two EF amplitudes, 7.5 V/mm (low) and 15 V/mm (moderate) were chosen based on our previous theoretical and experimental studies of the effects of EF on the cell membrane and cell responses in vitro [34,35]. Importantly, this range is well below electroporation voltage [36,37] and is known to be safe [36].

### 2.3. Flow Cytometry Analysis

For PS measurements, indicated cell lines were stained with annexin V-FITC (BD Bioscience, Franklin Lakes, NJ, USA) and propidium iodide (PI; Invitrogen, Waltham, MA, USA) according to the manufacturer’s protocols. Briefly, 1 × 10^6^ cells were incubated with annexin V binding buffer (BD Bioscience, Franklin Lakes, NJ, USA) and PI for 30 min at room temperature. Cells were washed with annexin V binding buffer and resuspended in this. For intracellular Ca^2+^ measurements, cells were stained with Fluo-3 fluorescein dye (Invitrogen, Waltham, MA, USA) according to the manufacturer’s instructions (Invitrogen, Waltham, MA, USA). For the membrane leakage test, cells were stained with YO-PRO-1 and PI dye according to the manufacturer’s instructions (Invitrogen, Waltham, MA, USA).

### 2.4. Immunofluorescence Staining

Cells were seeded on a petri dish containing gelatin (0.01%) coated coverslip. Upon reaching ~70% confluency, cells were washed twice with PBS (Corning, Manassas, VA, USA) and fixed with formaldehyde (Electron Microscopy Sciences, Hatfield, PA, USA) (4%) for 10 min at room temperature. After fixation, cells were washed twice with PBS and permeabilized with 0.1% Triton x-100 (Fisher Scientific, Waltham, MA, USA) for 5 min and washed twice with PBS and (1) for F-actin staining, cells were incubated with 100 nM Anti-actin 488 phalloidin (Cytoskeleton Inc., Denver, CO, USA) at room temperature in the dark for 30 min, for P38 MAPK staining, cells were stained with the primary anti-p38 MAPK antibody at a 1:1000 dilution (Abcam, Cambridge, UK) followed by anti-rabbit IgG (H+L), F(ab’)2 fragment Alexa Fluor 555 conjugated at 1:1000 (Cell Signaling Technologies, Danvers, MA, USA). The cells were washed twice with PBS. After washing, the coverslips were finally mounted using the anti-fade reagent Fluoro-gel II with DAPI (Electron Microscopy Sciences, Hatfield, PA, USA). Samples were analyzed using a BX51 fluorescence microscope with appropriate filters (Olympus, Tokyo, Japan) and a Biotek Lionheart FX (Biotek, Winooski, VT, USA) digital microscope.

### 2.5. Cell Cycle Arrest and Synchronization

Cells were grown to 70% confluency and then treated with hydroxyurea (HU; Sigma, USA) in complete medium to a final concentration of 4 mM for 16–24 h to synchronize the cells. Cell synchrony is needed for cell cycle studies because it enables the collection of population-wide data rather than relying solely on single-cell experiments [38].

Following synchronization, cells were incubated with a HU-free medium for 6 h to collect cells in the G1 phase. EF was applied to cells in the G1 phase for 6 h. As a control, cells were released for 6 h with a HU-free medium. The cell cycle stage was assessed via two standard methods, including Hoechst 33342 dye staining (Thermofisher, Waltham, MA, USA) and expression of cyclin D using cell lysates and flow cytometry [39,40].

### 2.6. SDS Page and Western Blots

For SDS-PAGE, 50 µg of whole cell lysates in RIPA buffer (Sigma-Aldrich, St. Louis, MO, USA) were denatured in SDS-loading dye (Bio-Rad, Hercules, CA, USA) and then loaded onto 4–15% denaturing gradient gels (Bio-Rad, Hercules, CA, USA). Proteins from the gel were transferred onto nitrocellulose membranes, and blots were blocked with 5% non-fat dry milk in PBS-0.1% Tween-20, followed by the addition of protein-specific antibodies and incubated overnight at 4 °C. The blots were washed with PBS-Tween-20 three times and incubated with HRP-coupled secondary antibodies. Following three washes with PBS-Tween-20, the blots were developed with Super Signal West Dura (Thermo Scientific, Rockford, IL, USA).

### 2.7. Statistical Analysis

One or two-factor ANOVA with Bonferroni post-hoc comparisons was used to determine the effects of EF on measured parameters. GraphPad software (GraphPad Prism, Version v6., Boston, MA, USA) was used to perform the statistical analysis. The results are reported as average ± standard deviation. A *p*-value of <0.05 was considered to denote statistical significance.

## 3. Results

### 3.1. Electric Field Stimulation Is Safe and Does Not Affect Cell Viability

A parallel plate capacitor device was used to generate the EF (Figure 1A). To demonstrate that there were no detrimental effects of EF on cell growth, viability, or membrane integrity, the following experiments were performed. EF of low (7.5 V/mm) or moderate amplitude (15 V/mm) was applied to glioblastoma cancer cells (U87-GBM) for six days. Cells were then counted using phase contrast microscopy to determine cell growth or stained with propidium iodide (PI) and counted using flow cytometry to quantify the percentage of dead cells. The results showed that there was no significant effect of the low (*p* = 0.5884, one-factor ANOVA, vs. control) and moderate EF (*p* = 0.0608, one-factor ANOVA, vs. control) stimulation on either cell growth (Figure 1B) or cell viability (Figure 1C, left and right). To demonstrate that EF stimulation does not cause membrane leakage (as occurs during electroporation), double PI and YO-PRO-1 staining was performed, followed by flow cytometric analysis. The results of the dot plots of PI/YO-PRO-1 showed that there is no significant difference between the percentage of double-positive YO-PRO-1 and PI cells between the EF-stimulated group (15 V/mm) and control cells, thus confirming no EF-associated membrane leakage (Appendix A).

### 3.2. Electric Field Modulates the Non-Apoptotic PS Exposure in Cancer Cells

To test if EF can be used to modulate surface PS, we applied EF stimulation of low and moderate amplitude to U87-GBM and cfPac-1 cancer cells or healthy astrocytes for 6 h. The results demonstrate that cell surface PS levels depend on the applied EF magnitude. Thus, while low amplitude EF stimulation resulted in a significant decrease in the surface PS levels (Figure 2A, top first and top second panels), moderate amplitude EF stimulation led to significantly greater surface PS levels in these cells (Figure 2A, bottom first and bottom second panels). Interestingly, there was no effect of either low or moderate EF stimuli on the surface PS expression in healthy astrocyte and HPDE cells (Figure 2A, top and bottom third panels). The shift in PS exposure in control and 6 h. low and moderate EF groups was evident in the flow cytometry histograms (Figure 2A, top and bottom fourth panels). For subsequent studies, we used U87-GBM cells because this cell line showed the most pronounced change in surface PS expression.

The exposure of phosphatidylserine on the outer leaflet of the cellular membrane is considered a unique feature of apoptosis, enabling the recognition and phagocytosis of dying cells [19]. However, PS presence is also reported in non-apoptotic forms of regulated inflammatory cell death [41]. To test whether the observed changes in PS surface expression were due to apoptosis-related mechanisms, we quantified cleaved Caspase 3 and Caspase 9 protein expression using Western blot analyses in the cancer cells (U87-GBM) and healthy cells (astrocytes) stimulated with both moderate and low amplitude for 24 h. Our results indicate that EF stimulation did not alter caspase-dependent apoptosis in healthy (astrocytes) or cancer (U87-GBM) cells, suggesting that EF-induced PS exposure in cancer cells occurs via caspase-independent mechanism (Figure 2B).

### 3.3. EF-Induced PS Exposure in Cancer Cells Regulate through Cytosolic Ca^2+^

PS externalization is generally regulated by increases in intracellular Ca^2+^ via inhibition of flippase activity [18]. It has been shown that modulation of the Ca^2+^ pathway by ionomycin or BAPTA-AM significantly changes the PS exposure on the outer cell membrane of cancer cells. On the one hand, ionomycin treatment increases the intracellular calcium influx into the cells, leading to increased surface PS levels in cancer cells by inhibiting the flippase activity. On the other hand, BAPTA-AM treatment significantly decreases the intracellular calcium level, which results in a slight decrease in the surface PS levels [18]. To determine if EF-induced PS exposure modulation may involve a similar mechanism, we quantified intracellular Ca^2+^ concentration on both healthy (astrocytes) and cancer (U87-GBM) cells following stimulation with low and moderate amplitude EF. As expected, EF stimulation did not affect Ca^2+^ concentration in healthy cells (Figure 3A). Interestingly, the EF stimulation of cancer cells results in differential regulation of cytosolic Ca^2+^ in these cells, with low amplitude EF leading to a significant decrease in cytosolic Ca^2+^ concentration, in contrast to a significant increase in cytosolic Ca^2+^ concentration following moderate amplitude EF stimulation (Figure 3A). The EF effect of cytosolic Ca^2+^ results correlate with the EF-induced PS exposure pattern that we observed (Figure 2A,B). To further elucidate the role of cytosolic Ca^2+^ on EF-induced PS exposure in these cells, we determined the effects of low and moderate amplitude EF stimulation on annexin V binding in Ca^2+^-free medium. We did not observe significant PS exposure change in both low and moderate amplitude stimulation groups once the cells were cultured with Ca^2+^-free medium (Figure 3B, left and right panels). These data suggest that EF-induced PS externalization is regulated through a Ca^2+^-dependent pathway.

### 3.4. Dual Effect of Electric Field on F-Actin Polymerization and Corresponding EF-Induced PS Exposure in Cancer Cells Provide Mechanistic Insight into EF-Mediated PS Regulation

Intracellular Ca^2+^ plays a critical role in the regulation of F-actin remodeling in muscle cells [42] and regulates actin cytoskeleton dynamics in cancer cells during cell migration [43]. Therefore, the following experiments were performed to determine if there is an association between actin polymerization and EF-induced PS exposure in cancer cells. First, we confirmed that actin polymerization in cancer cells leads to PS externalization, where cfPac-1, PANC-1 pancreatic cancer cell lines, and HPDE cells were treated with jasplakinolide and surface PS expression in the cells was measured. The results demonstrated that, as expected, stabilization of F-actin leads to PS externalization in these cells (Appendix A). To demonstrate the association between EF-induced PS modulation and actin polymerization, cancer cells (U87-GBM) were stimulated with low and moderate amplitude EF for up to 6 h, and the signal intensity of F-actin in these cells was quantified. We observed that low amplitude EF stimulation results in significantly decreased actin polymerization (*p* < 0.0001, one-factor ANOVA) (Figure 4A, upper right panel), in contrast to the moderate amplitude EF stimulation, which results in greater F-actin levels (*p* < 0.0001, one-factor ANOVA) (Figure 4A, upper right panel). We further examined whether EF changes actin polymerization in a time-dependent manner. We observed that low amplitude significantly decreased actin polymerization after 4 h (*p* = 0.0025, one-factor ANOVA) (Figure 4B, top left). Interestingly, we observed a significant increase in actin polymerization of cancer cells treated with 4 h moderate amplitude (*p* = 0.0065, one-factor ANOVA) (Figure 4B, bottom left). Furthermore, we found a trend toward a positive relationship between F-actin and PS exposure in cancer cells that were exposed to both low and moderate EF amplitudes, respectively (r^2^ = 0.9085, *p* = 0.0121; r^2^ = 0.7615, *p* = 0.0535) (Figure 4B, top and bottom right). Together, these results suggest that actin polymerization is involved in the regulation of EF-induced PS externalization in cancer cells.

### 3.5. Electric Field Has a Dual Effect on the Activation of p38 MAPK in Cancer Cells

Oxidative stress (OS) activates p38 mitogen-activated protein kinase (MAPK) pathways [44]. Oxidative stress has been shown to induce actin polymerization through the p38 MAPK/heat shock protein 27 (Hsp27) pathway in vascular endothelial cells [45]. To test the involvement of p38 MAPK and actin polymerization in cancer cells, we co-stained pancreatic cancer cells (MiaPaCa-2) with anti-phospho-p38 MAPK monoclonal antibody (mAb), and Alexa Fluor™ 488 coupled Phalloidin and imaged the cells using fluorescence microscopy. We observed that induction of oxidative stress triggered the translocation of phospho-p38 MAPK into the nucleus (Appendix A). This effect was not observed in the DMEM control, where phospho-p38 MAPK was co-localized with F-actin (phalloidin).

Intracellular Ca^2+^ plays a critical role in the regulation of p38 MAPK in cells [46]. It has been shown that an increased intracellular Ca^2+^ in PC-14 lung cancer cells following Dihydroartemisinin (DHA) treatment leads to activation of the p38 MAPK pathway. In contrast, a decrease in intracellular Ca^2+^ following BAPTA-AM treatment causes a reduction in the p38 MAPK activation [47]. Following our observation of the dual effect of EF on the intracellular Ca^2+^ concentration in cancer cells (Figure 3A), we investigated if this effect involves p38 MAPK activation using Western blot analyses. We observed that both low and moderate amplitude EF stimulation for 24 h increases p38 MAPK protein expression in cancer cells (U87-GBM) (Figure 5, top right panel). At the same time, low amplitude EF results in a decreased phospho-p38, and moderate amplitude EF results in an increased phospho-p38 at 24 h (*p* = 0.0407, low vs. moderate amplitude, one-factor ANOVA) (Figure 5, top right panel). Time-dependent studies show that p38 expression initially decreases (by 6 h) and then increases (by 24–36 h), with virtually undetectable levels of phospho-p38, following low amplitude EF stimulation (Figure 5, bottom left panel). There are no detectable changes in the protein expression of p44/42 ERK1/2 and AKT pathways in these cancer cells (Figure 5, bottom left panel). In contrast, moderate amplitude 24-h EF stimulation of the cancer cells (U87-GBM) increased the phospho-p38 MAPK protein expression (at 12, 24, and 36 h), and a slight increase in the p38 protein expression at 24 h, while again, no detectable changes in the protein expression of p44/42 ERK1/2 and AKT pathways are observed (Figure 5, bottom right panel).

### 3.6. Electric Field Stimulation Leads to Cell Cycle Arrest in Cancer Cells

Previous studies have shown that p38 MAPK regulates the G2/M and G1/S cell cycle checkpoints in response to cellular stress, such as DNA damage [48]. The p38 MAPK is activated by phosphorylation in response to DNA damage and leads to the establishment of a G2/M cell cycle checkpoint [48]. To determine whether EF-induced p38 MAPK activation can lead to cell cycle arrest, we synchronized cancer cells (U87-GBM) in the G1 cell cycle phase and then stimulated them with moderate amplitude EF for 24 h. Cell synchrony is needed for cell cycle studies since it allows us to collect population-wide data rather than relying solely on single-cell experiments. The results suggest that this stimulation simultaneously significantly increases the G2/M cell population (*p* = 0.0316, one-factor ANOVA) and decreases the G1 cell population (*p* = 0.0243, one-factor ANOVA), indicating that the moderate amplitude EF causes cell cycle arrest in the G2/M phase (Figure 6, left panel). Furthermore, this cell cycle arrest following EF stimulation is accompanied by a decreased protein expression of cyclin D1 (Figure 6, right panel), which is known to be predominantly expressed in the G1 phase of the cell cycle.

## 4. Discussion

PS is a well-established cancer biomarker, and it also plays the critical role in the cancer cell response to certain drugs [27,49,50]. In this study, we demonstrate that cell stimulation with non-contact electric field (EF) can be used to differentially modulate the surface PS expression of cancer cells. Our results show (Figure 7A) that EF stimulation has a dual effect on PS exposure on the cancer cell membrane depending on the EF amplitude and that these effects are mediated via the regulation of the intracellular Ca^2+^ levels, actin polymerization, p38 MAPK activation, as well as changes in the cell cycle. The potential mechanism for the EF effects on cancer cells is presented in Figure 7A. We propose that EF stimulation alters the membrane potential of the cells [35], which changes the activity of Voltage-Gated Calcium Channels (VGCCs) on the cancer cell membrane. The differential activation of VGCCs by EF of low or moderate amplitude causes different effects on intracellular Ca^2+^, actin polymerization, p38 MAPK activation, and cell cycle progression (Figure 7A). Our previous studies showed that the cancer cells, which have lower surface PS, are predominately in the G1 cell phase, and chemo and radiation treatments selectively target the lower surface PS cancer cells [20,27]. We also showed that the cancer cells with higher surface PS are generally in the G2/M cell phase, and SapC-DOPS selectively targets higher surface PS cancer cells [50]. Thus, a combination of moderate amplitude EF treatment with SapC-DOPS or low amplitude EF treatment with chemo/radiation may lead to enhanced cancer cell death, the main goal of anti-cancer therapy (Figure 7B).

Our findings raise an important question regarding how EF regulates PS externalization in cancer cells. Our previous theoretical study described the effects of the applied EF on live cells using realistic models of cell and cell membrane for a cell attached to a substrate [35]. In that study, we demonstrated that an external electric field causes significant surface charge redistribution and location-dependent fields to appear in the cell membrane and cytoplasm. Therefore, EF can be used to manipulate a variety of cellular responses that are mediated by ion channels, membrane receptors, etc. In particular, EF-induced changes in the membrane potential can alter the voltage-gated channels embedded in the cell membrane [51,52] and activate subsequent signaling. Therefore, the present study is the next in vitro step with a particular focus on the effects of EF on cancer cell surface biomarker modulation for potential translational therapeutic applications. Interestingly, our results suggest that moderate EF stimulation does not disturb the lipid membrane integrity (Appendix A) but causes transient PS exposure [1,50].

Ca^2+^ is a key regulator of PS exposure in cells as it regulates the activity of flippase enzymes, which are responsible for PS transfer between inner and outer cell membranes [18]. Our observation that EF regulates cytosolic Ca^2+^ levels suggests that EF-induced PS exposure is mediated by a Ca^2+^-dependent mechanism. EF-induced PS exposure was rescued once the cells were cultivated in Ca^2+^ free medium. These results indicate that under moderate amplitude EF stimulation, the increase in intracellular Ca^2+^ comes from the extracellular matrix and not from intracellular stores. This further suggests that plasma membrane channels are primarily involved, not ER channels. Other studies have indicated that electromagnetic fields and EFs increase the activity of voltage-gated Ca^2+^ channels in neural stem cells and chromaffin cells [53,54]. Interestingly, it has recently been reported that low-frequency electromagnetic fields increase Ca^2+^ channel expression in the presynaptic nerve terminal [55]. We, therefore, hypothesize that consistent with our previous work [35], EF alters voltage-gated channel activity or expression in cancer cells, thereby regulating cytosolic Ca^2+^ change.

It has been shown that different types of VGCC (Voltage-gated calcium channels), including L (long-lasting), P (Purkinje), N (neural), T (transient), and R (residual) types are activated under EF exposure. Each channel requires different depolarization voltages to be activated [56]. Future studies investigating the expression and types of Ca^2+^ channels that may be activated during EF would be beneficial in further understanding the mechanisms by which EF leads to PS externalization in cancer cells. In addition, testing whether EF modulates total PS expressed in cancer cells, including inner and outer leaflets on cancer cells, would provide insight into the nature of the increase in surface PS expression.

A limitation of this study is that it is conducted in vitro. Indeed, for potential therapeutic development, it is essential that the studies on whether EF induces PS externalization in cancer cells and enhances the efficacy of current PS targeting treatment need to be conducted ex vivo and in vivo. For in vivo testing, a device able to effectively apply EF to the tumor region will need to be designed. One of the possible issues for in vivo studies using the GBM mouse model is that the skull has a much lower conductivity than other tissues. Therefore, this should be considered while designing electrodes for in vivo studies.

While the activation of p38 MAPK upon induction of stress factors (e.g., oxidative stress) or active compounds (e.g., curcumin) has been reported previously [45,57], the role of p38 MAPK activation in PS exposure in cancer cells has not been demonstrated. Our results indicate a possible association between EF-induced PS exposure and p38 MAPK activation. This novel finding is intriguing and is consistent with the previous reports of the electric field effect on the activation of the p38-MAPK pathway [34,58]. Furthermore, our findings showed that even though EF modulates the activation of the p38 MAPK pathway, caspase-dependent apoptosis is not induced after 24 h EF induction in cancer (U87-GBM) or healthy (astrocytes) cells. At the same time, the activation of P38 MAPK with curcumin did induce caspase-dependent apoptosis in human breast cancer cells [57]. Overall, further studies of the effects of EF in cancer cells in the presence or absence of p38 MAPK inhibitors will be needed to determine the exact mechanisms for p38 MAPK in the externalization of PS in cancer cells in the context of EFs.

Previous studies suggest that PS expressed on the inner leaflet of the plasma membrane bilayer is connected to actin filaments and that actin helps with the clustering of lipids as well as transmembrane proteins on the cell membrane [59]. We found a positive correlation between actin polymerization and EF-induced PS exposure on the outer leaflet of cancer cells, which also supports the literature findings. However, the calcium pathway’s blockage effect on EF-induced PS exposure needs to be further examined.

## 5. Conclusions

To our knowledge, the work presented here represents the first mechanistic study of EF on the modulation of PS exposure in cancer cells. Our findings demonstrate that EF modulates PS exposure by altering the cytosolic calcium concentration. Cytosolic calcium regulates PS exposure on the cell surface of the cancer cells. EF-induced cytosolic calcium further modulates the P38 MAPK activity and leads to cell cycle regulation. Moreover, our findings demonstrate that EF induces PS exposure through Ca^2+^-modulated p38 MAPK-actin pathways and provides a rationale for a novel potential therapeutic strategy to enhance the efficacy of PS targeting cancer treatments. It has been shown that low and high PS cancer cells are more sensitive to chemo/radiotherapy and SapC-DOPS, respectively [26,27]. Indeed, altering the PS exposure with electric fields may enhance the treatment efficacy of these PS-targeting treatments. In the future, this novel EF treatment must be evaluated in vivo and in clinic with different types of tumors.

## 6. Patents

A patent (US 2021/0106819) was filed resulting from the work reported in this manuscript.

## Figures and Tables

**Figure 1 biomedicines-11-00466-f001:**
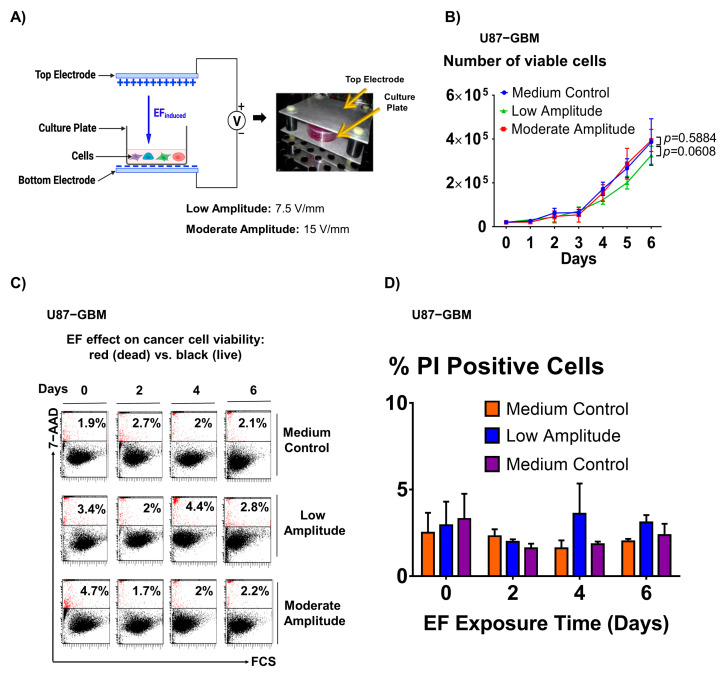
Electric field stimulation is safe and does not cause detrimental effects on cell growth, viability, and membrane integrity. (**A**) In vitro electric field stimulation setup. (**B**) There is no difference in the number of viable glioblastoma (U87-GBM) cells following EF stimulation and the medium control group. Sample size (n) = 5. Flow cytometry dot plots (**C**) and a bar graph (**D**) demonstrate that there is no difference in the percentage of PI-positive cells between EF-stimulated and control groups. Sample size (n) = 3.

**Figure 2 biomedicines-11-00466-f002:**
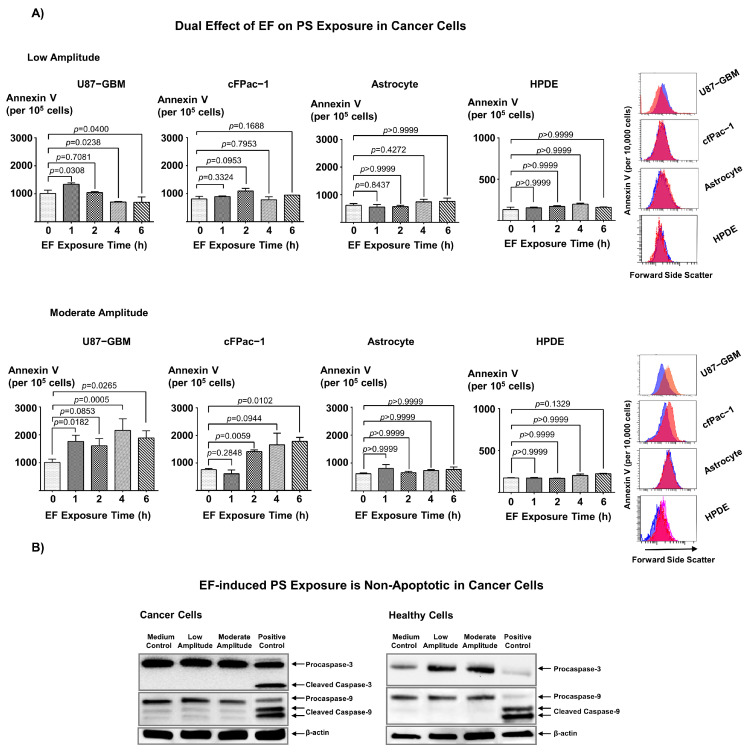
Electric field stimulation induces PS exposure in cancer cells via caspase-independent mechanism. (**A**) Flow cytometric measurements of annexin V (FITC Fluorescence) demonstrate a significant decrease or no change in cell PS exposure in U87-GBM (top first panel) and cfpac-1 (top second panel) cancer cells following low amplitude EF (top panel) for 4–6 h, whereas moderate amplitude EF (bottom panels) stimulation significantly increased the PS exposure in U87-GBM (bottom first panel) and cfpac-1 (bottom second panel). This effect is absent in healthy astrocytes (top third, bottom third panels) and HPDE cells (top fourth, bottom fourth panels). The top and bottom fifth panels show representative histograms, with blue denoting Control and red indicating the EF-treated groups (6 h). Sample size (n) = 3 (**B**) Western blot analysis showed that 24-h EF stimulation did not change cleaved caspase 3 and 9 expressions in glioblastoma cells (U87-GBM) (left panel) or healthy cells (astrocytes) (right panel), with the levels in the stimulated group similar to the controls. Positive control: Cells were treated with 1 μM staurosporine for 4 h.

**Figure 3 biomedicines-11-00466-f003:**
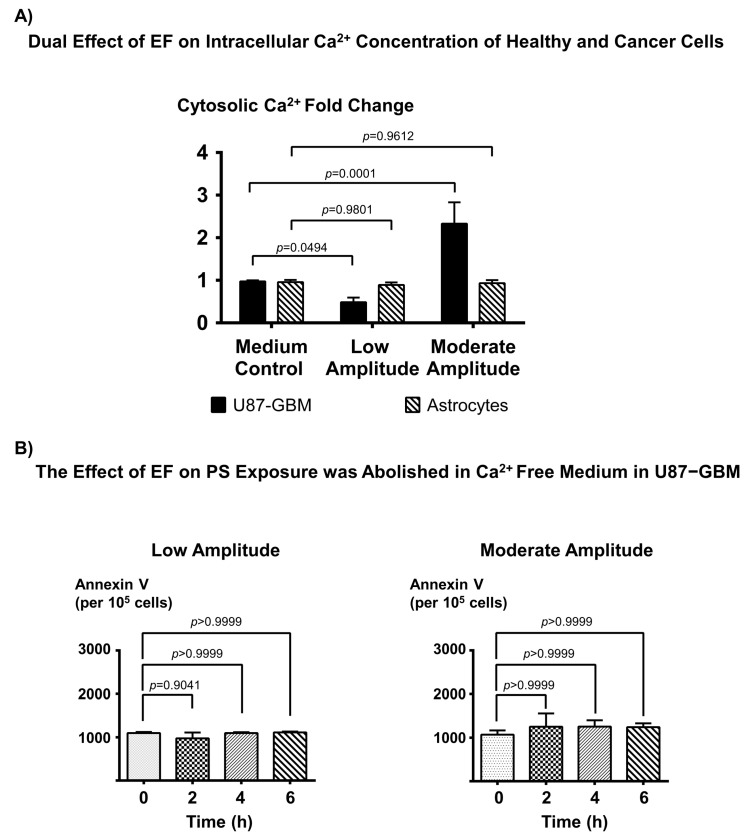
The electric field differentially regulates the intracellular Ca^2+^ levels in cancer cells. (**A**) Intracellular Ca^2+^ concentration (Fluo-3) fold changes of glioblastoma cells (U87-GBM) and healthy cells (astrocytes) cultured in normal medium (DMEM) following low and moderate amplitude EF stimulation for 6 h. The results show that low-amplitude EF stimulation decreases intracellular Ca^2+^ concentration, while moderate-amplitude EF stimulation increases intracellular Ca^2+^ concentration in cancer cells. Sample size (n) = 3 (**B**) Flow cytometric measurements of annexin V (FITC Fluorescence/10^5^ cells) on glioblastoma (U87-GBM) cells following low and moderate amplitude EF stimulation in the Ca^2+^-free medium. The results show that the EF-induced PS exposure in cancer cells, as shown in (**A**)**,** is completely abolished in the absence of Ca^2+^, suggesting the critical role of Ca^2+^-dependent mechanisms in the EF-mediated change in PS exposure in cancer cells. Sample size (n) = 3.

**Figure 4 biomedicines-11-00466-f004:**
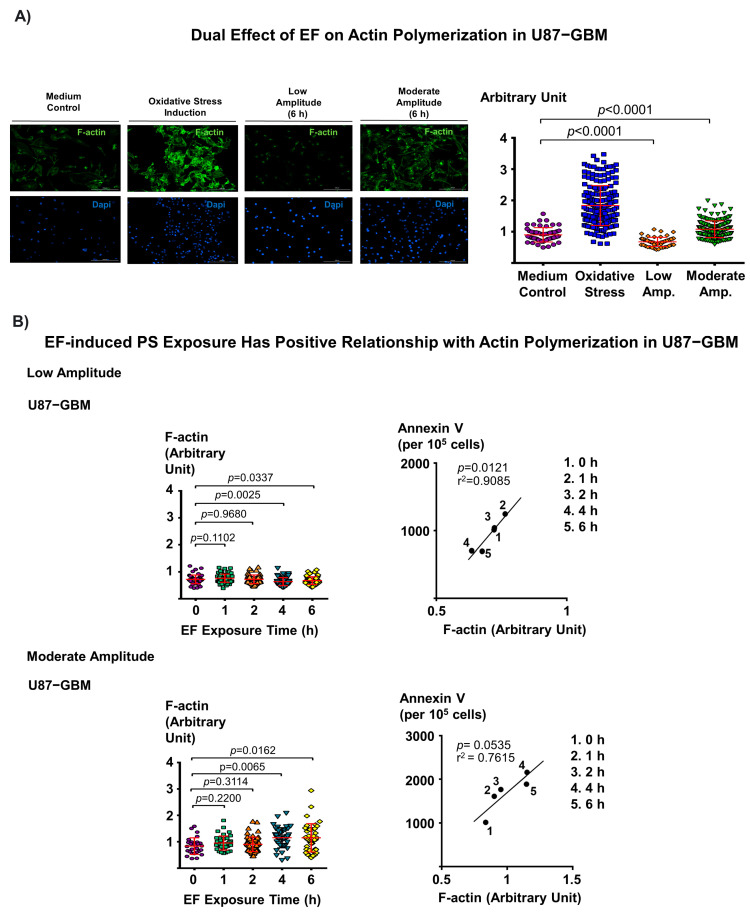
Actin polymerization is involved in the regulation of EF-induced PS externalization in cancer cells. (**A**) Representative immunofluorescence images of glioblastoma cells (U87-GBM; per 10^5^ cells) (left panel) demonstrating that low-amplitude EF stimulation decreases actin polymerization, while moderate-amplitude EF stimulation increases actin polymerization levels in cancer cells. Negative control: culture medium; positive control: oxidative stress induction with 1.2 mM H_2_O_2_ for 20 min. (**B**) There is a trend toward a positive relationship between F-actin and PS exposure of cancer cells in both low and moderate amplitude EF stimulation groups (r^2^ = 0.9085, *p* = 0.0121 vs. r^2^ = 0.7615, *p* = 0.0535, low vs. moderate amplitude, respectively).

**Figure 5 biomedicines-11-00466-f005:**
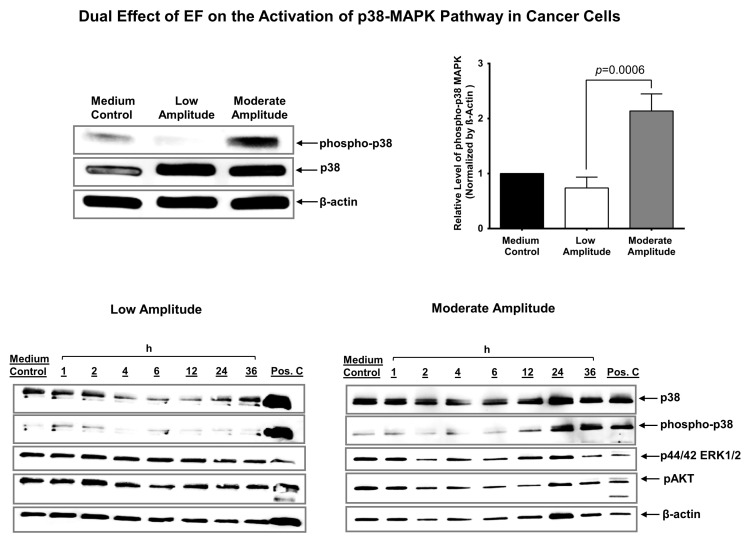
Electric field modulates the p38 MAPK activity in cancer cells. Western blot analysis showing the expression of p38 MAPK, phospho-p38 MAPK, and beta-actin of glioblastoma cells (U87-GBM) treated with DMEM, low and moderate amplitude EF for 24 h (top left panel). Western blot analysis of relative phospho-p38 MAPK protein levels between medium control, low and moderate amplitude groups (top right panel). Sample size (n) = 3. Western blot analysis showing the expression of p38 MAPK, phospho-p38 MAPK, P44/42 ERK1/2, p-AKT, and beta-actin of glioblastoma cells (U87-GBM) treated with DMEM (control) and low amplitude EF for indicated time periods (bottom left panel). Western blot analysis showing the expression of p38 MAPK, phospho-p38 MAPK, P44/42 MAPK, p-AKT, and beta-actin of glioblastoma cells (U87-GBM) treated with DMEM (control) or moderate amplitude EF for indicating time periods (bottom right panel).

**Figure 6 biomedicines-11-00466-f006:**
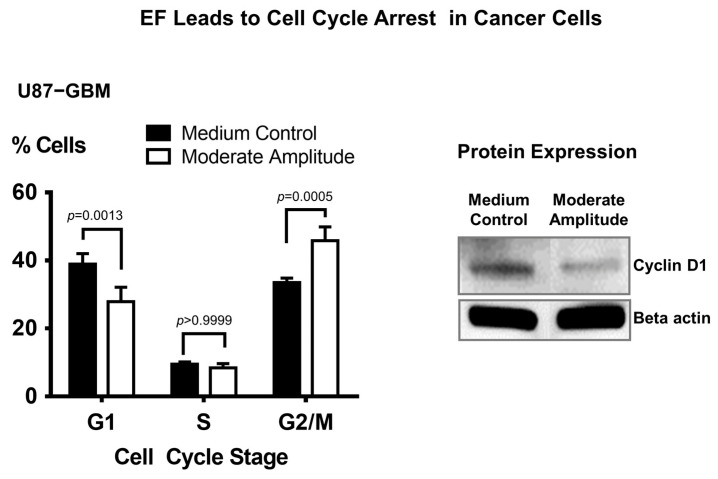
Moderate-amplitude electric field stimulation causes cell cycle arrest in glioblastoma cancer cells. Following EF stimulation of glioblastoma cells (U87-GBM) for 24 h, there was a significant change in the cell cycle distribution pattern (left panel), with lower percentage of cells in G1 and higher percentage of cells in G2/M phases in the EF groups as compared to the respective percentages in the control groups. Western blot analyses demonstrate that 24 h stimulation with moderate amplitude EF results in a decreased protein expression of cyclin D1 in glioblastoma cells (U87-GBM) (right panel). Sample size (n) = 3.

**Figure 7 biomedicines-11-00466-f007:**
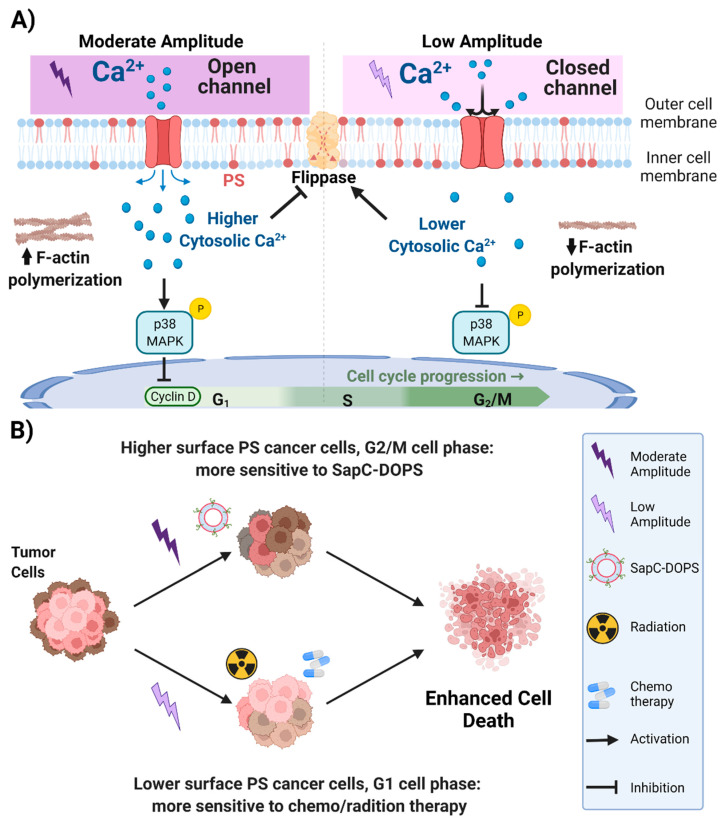
(**A**) Possible mechanism of action of the electric field in cancer cells. Moderate amplitude EF activates the voltage-gated calcium channels on the cancer cell membrane, leading to cytosolic calcium increase within the cells. Increased cytosolic calcium inhibits flippase activity and further accumulates PS on the outer cell membrane. Next, higher cytosolic calcium induces actin polymerization and activates the p38 MAPK pathway in cancer cells; this activation blocks the expression of cyclin D. On the other hand, low amplitude EF inhibits the activation of calcium channels, which leads to a lower cytosolic calcium concentration within the cells. Decreased levels of cytosolic calcium results in the activation of the flippase enzymes, which leads to reduced PS localization on the outer cell membrane. Moreover, this process inhibits further activation of p38 MAPK and actin polymerization. (**B**) Potential therapeutic application of low amplitude electric field. Moderate amplitude EF increases the PS exposure of cancer cells, whereas low amplitude EF decreases their PS exposure. The cancer cells treated with the moderate amplitude EF will end up with higher PS exposure and become more sensitive to high-PS targeting treatments such as SapC-DOPS. Conversely, cancer cells treated with low amplitude EF will have lower PS surface exposure and, therefore, become more sensitive to radiation and chemotherapy. Thus, the combination of moderate amplitude EF treatment with SapC-DOPS or low amplitude EF treatment with chemo/radiation may lead to enhanced cancer cell death, the main goal of anti-cancer therapy.

## Data Availability

Not applicable.

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
