# Peer review of "Electric Fields Regulate In Vitro Surface Phosphatidylserine Exposure of Cancer Cells via a Calcium-Dependent Pathway"

_biomedicines, 2023, doi:10.3390/biomedicines11020466_

Round 1

Reviewer 1 Report

The authors reported electric field modulated phosphatidylserine (PS) exposure on cancer cells and studied its underlying mechanisms.

The manuscript is overall well-structured.  Methods and experiments are described with sufficient information. However, there are some issues in terms of study designs and conclusions

1, In the Introduction, there is one misstatement, “Irreversible electroporation has been used in preclinical and clinical studies with a variety aims including: the induction of cytoskeleton instability [2], delivering anti-cancer agents or molecules into cancer cells [3,4], and transfection of cells [5].” It is the reversible electroporation that is used for the drug delivery (electrochemotherapy, ECT) and DNA/RNA delivery (Gene electro-transfer, GET) whereas IRE is used for tumor ablation. The cell death induced by IRE differs mechanistically from the tumor clearance by ECT or GET.

2, In many experiments (Figs. 1, 2A, 6), n=2 was adopted. How did the authors get all these statistical significances?

3, So far, experiments only demonstrated that PS exposure is external calcium dependent. However, whether other downstream signal pathways, such as actin polymerization, p38MAPK activation and cell arrest are dependent on external calcium and how this pathway/change directly triggers PS exposure is not substantially confirmed. The impact of blockage of calcium signal on the p38 MAP pathway or the blockage of this pathway on PS exposure are not experimentally validated. Thus, part of conclusion is largely the speculation.

4, In the Fig. 1c, moderate EF appears to increase the cell death at day 4 and possible at day 6 as well. Additionally, Fig. 2 showed two types of cancer cells responded to the low EF differently. One reduced the PS exposure but the other did not change it.

One minor issue is the writing consistence. For example, “26 mm” in Line 105 but ”7.5V/mm (low) and 15V/mm” in Line 109. Sometime there is a space but sometime there is no space between the number and the unit.

Another minor issue is no Fig. legends provided in the Supplemental Figures.

Reviewer 2 Report

In general, the manuscript makes a good impression.

In section 3.6 please explain why it was necessary to use synchronized cancer cells in this experiment.

Reviewer 3 Report

Specific comments to the authors

The authors Ahmet Kaynak et al. of the submitted manuscript „Regulation of Phosphatidylserine Exposure on the Surface of Cancer cells by Electric Fields thorough a Calcium Pathway” studied the possible effects of novel non-contact electric fields (EF) inducer device on cancer biomarkers in normal and cancer cell lines (in detail glioblastoma cell line (U87-GBM) and human pancreatic cancer cell lines (cfPac-1 cells and MiaPaCa-2) as well as the human astrocyte and pancreatic ductal epithelial (HPDE) cells). Based on the applied molecular techniques (flow cytometry analysis, immunofluorescence staining, Western blots) the authors could demonstrate that EF can regulate cell surface phosphatidylserine (PS) and the intracellular calcium (Ca2+) levels of the used human normal control and cancer cell lines. Furthermore, EF could induced the p38 mitogen-activated protein kinase (MAPK), actin polymerization, and cell cycle arrest in the used cancer cell lines, too. Therefore, the authors of the study suggested postulated that EF targets surface PS in cancer cells in vitro, which could be therefore applied as a biomarker for EF.

Overall, the manuscript give some interesting insights of possible mechanistic of EF on human tumor cell lines in vitro. Although the manuscript (including presentation) is mostly comprehensible and convincing, the different application of the human normal and cancer cell lines is largely confusing. The methods are mostly well described. Although the results and discussion are clearly presented, the major concerns relate to the missing transfer to “real” in-vivo or in-situ situation. Therefore, the authors (see specific comments) must perform some minor to major changes to improve the manuscript. In conclusion, the presented data are interesting. After incorporating the mentioned specific comments (see below) the manuscript has the potency to be accepted.

Specific comments

Title: The title should indicate the in-vitro-character of the presented study. Please change adequately.

Abstract: Please specify the used human normal control and cancer cell lines in the abstract. The conclusion of the abstract makes no sense, since all investigations of the presented study based on in-vitro experiments.

Introduction: Please indicate the content-related differences between the presented study and the publication under reference number 28.

Material&Methods: It is not clear why this type of human control and cancer cell lines are selected. Please explain. Please specify the type of post-hoc comparison (such as Bonferrroni, Tukey, Duncan…)

Results:

# Figure 1: What is the reason for the cut-of-value of 7.5 and 15 V/mm. Please specify. Please indicate the used human cell line in Fig1C and Fig1D.

# Figure 2: The results of the human “control” pancreatic cell line are missing in the figure.

# Figure 3: Is the sample size “(n)=2” in figure 3A correct? If yes, the sample size is not sufficient for the applied statistical analysis.

# Figure 5: It is not clear, why the pancreatic cancer cell line MiaPaCa-2 was used for the experiment dealing with p38/MAPK-pathway? Please explain.

# Figure 6: Another marker of proliferation/mitotic activity (like Ki67 or pHH3) should be used to support the findings (“EF leads to Cell Cycle Arrest in Cancer Cells”).

Discussion: Please indicate the content-related differences between the presented study and the publication under reference number 41. The limitation of the presented study should be mentioned (no in-vivo or in-situ experiments). How could the interesting findings transferred from a theoretical to a practical view and especially clinical aspect? Please discuss in short.

Reviewer 4 Report

1) For cell cultures, it should be indicated from which passage the cells were used in the experiments. How were mycoplasma tests performed? It would be nice to have pictures of the cell cultures that were used for the experiments.

2) It seems to me that for figure 2 the number of samples should be increased to at least three. % PI positive cells should be designated D.

3) Legend for figure 3 should be more informative

4) It would be nice if the authors did experiments with Voltage-Gated Calcium Channels blockers

5) The excellent work presented needs a more detailed conclussion to reflect the full scope of the results obtained.

Round 2

Reviewer 3 Report

Specific comments to the authors

In the revised version of the manuscript, the authors have addressed the previously mentioned concerns in a very adequate and convincing manner. All points of criticism are answered by additional explanations or experiments. Therefore, I suggest that the revised manuscript "Electric Fields Regulate In Vitro Surface Phosphatidylserine Exposure of Cancer Cells via A Calcium-Dependent Pathway" should be accepted.

Reviewer 4 Report

All my comments have been taken into account. The article can be accepted for publication in its current form